Detailed morphological characterization and improvement of keratinocyte outgrowth from plucked human hair follicle

Klingenstein Stefanie stefanie.klingenstein@uni-tuebingen.de
Boenke Judith
http://orcid.org/0000-0002-8850-2317 Wüstner Lisa-Sophie
Liebau Stefan
Klingenstein Moritz
Institute of Neuroanatomy and Developmental Biology (INDB), Eberhard Karls University Tübingen , Tübingen , Germany
Gould Gwyn
Electronic publication date: 2025 Oct 31
Publication date: 2025
Volume: 13
Electronic Location ID: e20214
Received 2025 Jun 5; Accepted 2025 Sep 19
Copyright: © 2025 Klingenstein et al.
Copyright year: 2025
Copyright holder: Klingenstein et al.
License: This is an open access article distributed under the terms of the Creative Commons Attribution License, which permits unrestricted use, distribution, reproduction and adaptation in any medium and for any purpose provided that it is properly attributed. For attribution, the original author(s), title, publication source (PeerJ) and either DOI or URL of the article must be cited.
License URL: https://creativecommons.org/licenses/by/4.0/

Keywords: Plucked hair follicle, Outer root sheath (ORS), Keratinocytes, iPSCs, Cytokeratins, Keratins, Keratocysts

Funding: DFG DFG LI 2044/5-1, DFG LI 2044/6-1 This study was funded by the DFG (DFG LI 2044/5-1, DFG LI 2044/6-1 to Stefan Liebau). The funders had no role in study design, data collection and analysis, decision to publish, or preparation of the manuscript.

==============================
In this study, a detailed analysis of the outgrowth of primary keratinocytes from plucked human hair follicles was conducted. Plucked hair follicles offer an easily accessible and non-invasive method as a primary cell source for adult somatic keratinocytes, providing a simple starting material for induced pluripotent stem cell (iPSC) reprogramming. In this study, we laid our focus on the precise examination of timing and location of the first keratinocyte outgrowth after plucking, as well as the morphological changes that occur during cultivation. Our results show that the region of the hair follicle, from which the initial outgrowth occurs, is crucial for successful cultivation. Additionally, first appearing protrusions and first visible cells show the same specific marker expression as the intact outer root sheath. We therefore conclude that initially appearing keratinocytes arise from the basal layer of the outer root sheath. To improve the whole process, the protocol was adjusted to reduce the time until successful outgrowth. These optimizations are particularly relevant for developing a standardized protocol that works efficiently across all individuals, including patients with difficult-to-culture hair follicles. The accelerated cell harvesting could prove valuable in future applications in regenerative medicine, especially for patients where culture times have previously been too long.

Introduction

Human hair is a thread-like structure composed of a protein-rich matrix. It plays a role in skin protection, thermoregulation, and sensory perception. Hair grows from hair follicles (HFs), which are embedded deep in the dermis. Keratinocytes are the dominant cells in the epidermis, making up about 90% of skin cells (McGrath, Eady & Pope, 2004). These cells are responsible for synthesizing keratins, an important group of structural proteins. In HFs, keratinocytes are mainly found in the outermost layer of the hair organ and the hair cortex, where they contribute to hair formation and stability. Keratinocytes undergo continuous differentiation, moving from the basal layer of the skin outward to form the outermost skin layer, which serves as a barrier to the environment (Houben, De Paepe & Rogiers, 2007). The HF is a complex mini-organ anchored deep in the dermis and consists of several layers (Wong et al., 2016). Beyond its role in hair production, the hair follicle is embedded in a highly dynamic skin environment, where it interacts with dermal fibroblasts, immune cells, vasculature, and the extracellular matrix. These interactions are crucial for processes such as hair cycling, wound healing, and immune modulation (Wong et al., 2016; Sasaki, 2019). As part of the cutaneous autoregulatory network, the follicle is both influenced by and contributes to local signaling environments, which regulate keratinocyte behavior and stem cell activity. The hair shaft is the visible part of the hair that extends beyond the skin surface. The hair papilla is the most proximal part of the hair organ, is supplied with blood vessels and responsible for hair growth. The matrix is the area of the HF where cell division and hair growth occur (Lin, Zhu & He, 2022). Different layers of the hair organ can be further distinguished: The inner root sheath helps shape and support the growing hair shaft during its development. The outer root sheath (ORS) extends from the epidermis to the hair papilla, surrounding both the inner root sheath and the hair itself. It contains keratinocytes and stem cells essential for HF regeneration and hair growth (Li et al., 2021).

Keratinocytes can be further characterized by specific proteins. These are fibrous structural proteins found not only in hair but also in skin and nails. They are synthesized by keratinocytes and are crucial for the strength and elasticity of hair (Langbein et al., 1999, 2001). There are two main types of keratins in hair. The soft keratins are found in the ORS and skin, and the hard keratins are found in hair and nails (Li et al., 2021). These proteins enable hair to withstand mechanical and environmental stressors and return to its original shape after stretching. While the use of ORS-derived cells from plucked HFs is well established (Weterings, Vermorken & Bloemendal†, 1981), detailed morphological studies addressing the exact outgrowth region, timing, and origin of the emerging keratinocytes are still lacking.

Plucked HFs, particularly ORS cells, provide, a non-invasive source of keratinocytes (Raab et al., 2014). Of particular interest is the use of these keratinocytes for generating induced pluripotent stem cells (iPSCs) (Klingenstein et al., 2020). iPSCs can differentiate into virtually any cell type in the body, offering enormous potential for regenerative medicine and disease research. By the forced introduction of specific transcription factors, like OCT3/4, SOX2, KLF4, and c-MYC, keratinocytes can be reverted to a pluripotent state (Al Abbar et al., 2020). Keratinocytes offer several advantages as a starting cell type for generating iPSCs. They are easily accessible, as they can be isolated in a non-invasive manner through HF harvesting, making the procedure less invasive and more comfortable for patients. They also exhibit a high reprogramming rate compared to other somatic cells, like mesenchymal stem cells or epidermal cells from urine, which makes the reprogramming process more efficient (Klingenstein et al., 2020). A key factor in reprogramming keratinocytes to iPSCs is the convenient way to realize a personalized therapy.

Since keratinocytes can be easily obtained from patients, they enable the development of personalized stem cells for disease-specific research approaches and regenerative therapies. Al though the use of plucked human hair follicles for iPSC generation has become increasingly popular due to its accessibility, detailed morphological studies of the early keratinocyte outgrowth process are still lacking. Most previous work has focused on the reprogramming outcome, while little attention has been given to the structural and temporal characteristics of cell emergence, particularly in relation to the ORS and its stem cell populations (Lin, Zhu & He, 2022; Li et al., 2021; Zhang & Chen, 2024).

With this study, we aim to fill this gap by providing a comprehensive, standardized morphological characterization of keratinocyte outgrowth from plucked human hair follicles, including the origin, timing, and marker expression of the first emerging cells. Our findings not only offer a better understanding of this commonly used but under-characterized cell source but also suggest improvements to existing protocols—potentially enhancing the efficiency of personalized iPSC generation in both research and clinical settings.

Materials and Methods

Plucking and standard cultivation of human HF

HFs from four different voluntary donors (two male, two female) were used in total. Detailed information about the donors can be found in Supplemental Table T1. Written informed consent was obtained from all participants prior to inclusion in the study. The study was conducted in accordance with the Declaration of Helsinki and approved by the Ethics Committee at the Medical Department of the University Tübingen (Project number: 638/2013BO01, date of approval: 31.3.2014). The human HFs were plucked from the parietal scalp region using tweezers and immediately transferred into Dulbecco’s Modified Eagle Medium (DMEM) transport medium (Thermo Fisher, Waltham, MA, USA). For standard cultivation, the HFs were plated in tissue culture dishes and initially cultured in Mouse Embryonic Fibroblast (MEF) medium. The detailed composition of the MEF medium is provided in Supplemental Table T2. This two-phase culture protocol was established in our group based on previous work (Wustner et al., 2022), showing that MEF medium during the initial 72 h promotes adhesion and activates keratinocytes within the ORS, which are initially in a quiescent state. In contrast, when cultures were started directly in EpiLife medium, keratinocyte outgrowth was significantly reduced or absent. Therefore, daily monitoring and morphological assessment were performed during MEF cultivation. Once the first keratinocytes were visible, the medium was switched to EpiLife Keratinocyte Medium with HKGS supplement (Thermo Fisher, Waltham, MA, USA) to support further proliferation. The detailed composition of the keratinocyte medium is provided in Supplemental Tables T2 and T3. For more detailed information on the cultivation of plucked human HFs, please refer to our earlier publication (Klingenstein et al., 2020).

In the experimental setup designed to shorten the time until the first keratinocytes appeared, specific cell culture supplements were added to the MEF medium. Detailed information on these supplements can be found in Supplemental Table T3.

Embedding and sectioning of plucked HF

After plucking, the HFs were trimmed 1–2 mm above the ORS, resulting in a piece approximately 5–6 mm long, primarily consisting of the HF itself. The HFs were then fixed in 4% phosphate-buffered paraformaldehyde (Santa Cruz, Dallas, TX, USA) for 20 min at room temperature. After washing with DPBS−/− (Thermo Fisher, Waltham, MA, USA), the samples were transferred into EDTA (Roth, Newport Beach, CA, USA) for 2–3 days to soften the keratinized hair shaft for sectioning. The HFs were embedded in Tissue-Tek O.C.T. (Sakura, Osaka, Japan) compound and sectioned using a Cryostat Microm HM560 (Thermo Fisher, Waltham, MA, USA) at a sample temperature of −21 °C and a blade temperature of −20 °C. The slices were cut to a thickness of 14 μm for both cross and longitudinal sections.

Histological staining of plucked HF

For the histological hematoxylin and eosin staining, the 14 µm thick sections were first rehydrated in DPBS−/− for 5 min and briefly rinsed in distilled water (Roth, Newport Beach, CA, USA). After a 30-s incubation in hematoxylin solution (Roth, Newport Beach, CA, USA), the sections were blued under running tap water. For counterstaining, the sections were incubated in eosin (Roth, Newport Beach, CA, USA) for 30 s. This was followed by a graded alcohol series (70%, 95%, 100% ethanol, 30 s each), and finally, the sections were cleared in xylene (Roth, Newport Beach, CA, USA) for 30 s and mounted with DPX (Sigma-Aldrich, St. Louis, MO, USA).

Immunofluorescence staining of plucked HF and keratinocytes

Before rehydration, HF sections were circled with a hydrophobic pen (DAKO) and immersed in an ethanol gradient (70% - 95% - 100% - 95% - 70% ethanol, 30 s each), followed by washing with DPBS−/−. The sections were then rehydrated in DPBS−/− for 5 min. Afterward, the slices were blocked with skimmed milk blocking buffer (Supplemental Table T4) for 45 min at room temperature. Primary antibodies, diluted in skimmed milk blocking solution (Supplemental Table T4), were applied to the samples, and incubated overnight at 4 °C. The following day, the sections were washed three times for 5 min each, and secondary antibodies diluted in DPBS−/− (Supplemental Table T4) were added. After 1 h of incubation at room temperature, protected from light, the sections were mounted with Mowiol (Roth, Newport Beach, CA, USA).

Microscopy

The brightfield images were all captured using the PrimoVert Axio Imager M2 microscope (Zeiss, Oberkochen, Germany). The histologically stained HF were imaged with the AxioScan Z1 (Zeiss, Oberkochen, Germany). The immunofluorescence images were taken using the Axio Imager M2 microscope with Apotome and Plan-APOCHROMAT 20x/0.8 objective. The images were analyzed using AxioVision 4.8.1 software and ZEN Blue (both Zeiss).

RNA extraction and quantitative gene expression analysis

RNA was purified using the RNeasy Mini/Micro Kit (Qiagen, Hilden, Germany). Samples were first washed with DPBS−/− and then resuspended in 350 μl of lysis buffer until complete homogenization was observed. The homogenized solution was transferred to a shredder column and centrifuged for 1 min at 11,000 × g. The column was discarded, and the flow-through was used for subsequent steps of RNA purification following the kit’s instructions. Finally, RNA was eluted with 16 μl of RNAse-free water. OneStep quantitative real-time PCR was conducted using a StepOne instrument (Applied Biosystems, Waltham, MA, USA) and the QuantiFast SYBR™ Green RT-PCR Kit (Thermo Fisher, Waltham, MA, USA), following the manufacturer’s protocol. Since the kit includes the reverse transcriptase enzyme, no separate cDNA synthesis was required, and the extracted RNA was used directly. The mRNA levels of K5, K10 and K14 were quantified and normalized to the mRNA level of the housekeeping gene GAPDH. The following probes from the company Thermo Fisher were used: GAPDH (#Hs99999905_m1), K5 (#Hs00361185_m1), K10 (#Hs00166289_m1), K14 (Hs00265033_m1).

Statistical analysis

For the analyses of keratinocyte outgrowth timing, morphological changes of plucked hair follicles, and keratin expression in outgrowing keratinocytes, data are presented as mean ± standard error of the mean (SEM). Group comparisons were performed using unpaired, two-tailed Student’s t-tests assuming equal variances. For the experiments testing the effects of different additives on keratinocyte outgrowth speed, data are presented as mean ± standard deviation (SD). Here, group comparisons were performed using unpaired, two-tailed Welch’s t-tests, accounting for unequal variances. A p-value < 0.05 was considered statistically significant. Sample sizes (n) refer to independent biological replicates and are stated in the corresponding figure legends.

Results

Anatomy and culture of human hair follicles

To harvest keratinocytes from plucked human HFs for reprogramming into keratinocyte-induced pluripotent stem cells (kiPSCs), it was essential to first thoroughly examine the structure of the hair organ and the HF. This helped us understand which HFs were suitable for keratinocyte isolation and why certain follicles were better suited for this purpose.

Human hair lies in the dermis, the middle part of the human skin and passes through three major growth phases (Fig. 1A) (Natarelli, Gahoonia & Sivamani, 2023). Most hair roots are in the growth, or anagen phase which can last 3 to 6 years (Paus & Cotsarelis, 1999). The anagen phase is followed by a transition to the catagen phase. Within a few days, the HF recedes and enters the telogen phase, which lasts for 3 to 4 months. During this resting phase, the club hair remains anchored in the follicle but is no longer actively growing (Alonso & Fuchs, 2006). To examine the hair organ anchored in the skin, a skin biopsy is usually required. However, since this involves a minor surgical procedure that requires medical staff, specialized equipment, local anesthesia, and aftercare, there is a simpler and painless method to study these structures. By simply plucking HF—something that naturally happens in everyday life, for instance, when combing the hair—HF in the anagen phase can be specifically identified and used for further investigations. Anagen hair roots are anchored in the dermis by the arrector pili muscle (Poblet, Ortega & Jiménez, 2002). The efferent nerve fibres, as well as blood vessels, sebaceous glands and sweat glands are embedded in the loose connective tissue of the dermis around the HFs (Fig. 1B) (Sasaki, 2019). In contrast to skin-embedded HFs, plucked hair lacks the arrector pili muscle, all types of glands, blood supply and nerval innervation. Also, the outermost dermal sheet is missing (Fig. 1C). In skin-embedded and plucked HF the ORS remains clearly visible and intact, indicating that the HF meets all the criteria necessary for successful keratinocyte isolation from the hair root (Fig. 1D). DAPI staining reveals the layer-by-layer structure of the HF revealing distinct cell layers and the characteristic convex shape of the human hair organ (Fig. 1E). As shown in Fig. 1F, telogen HFs lack the morphological integrity of the ORS, which is essential for keratinocyte outgrowth. Keratinocytes are the major cell source found within the HF and they are arranged in characteristic structures, represented by different densely packed clusters and layers around the middle hair shaft (Lin, Zhu & He, 2022; Martel et al., 2024). Primary keratinocytes can be isolated from HFs in the anagen phase. During this phase, the cells, particularly in the ORS and other parts of the HF, are highly proliferative, metabolically active, and the follicle remains structurally intact and fully developed (Figs. 1F, 1G) (Klingenstein et al., 2020). The ORS contains a higher number of stem cells in the anagen phase, which can easily differentiate into keratinocytes, making the isolation and cultivation of these cells more efficient (Aasen & Belmonte, 2010). Magnified view of the boxed region in Fig. 1G, showing the first primary keratinocytes migrating out from the ORS of the plucked anagen hair follicle (Fig. 1H). But these stem cells are not only essential for hair growth; they also play a crucial role in differentiating into specialized cell types, such as glandular cells, in the repair of skin injuries, and in maintaining the function of the hair follicle (Ji et al., 2021). In other phases, particularly the telogen phase, when the HF is in a resting state the cells are not actively dividing and the ORS is less accessible (Tung & Yasuda, 2020). To successfully isolate primary, somatic keratinocytes from HFs, a specific cultivation protocol is required (Fig. 1I). For this, HFs were plucked from the parietal region of the scalp and cultivated adherently in tissue culture plates with MEF medium supplemented with growth factors. After approximately 72 hours (h), the first keratinocytes began to grow out from the HFs, and the medium was then switched to EpiLife with HKGS supplements. The keratinocytes were allowed to proliferate until they reached a sufficient density, after which they were further processed.

Figure 1 Anatomy and culture of human hair follicles (HFs) and study design.

The cell cycle of human HFs includes the three stages of anagen, catagen, and telogen hair growth (A). Histological H&E staining directly compares a skin-embedded HF (B) with a plucked HF (C), highlighting the clearly visible outer root sheath (ORS). With the inverted DAPI staining (D) the ORS, IRS and basal layer of the ORS are clearly visible. A cross-section of a plucked HF shows DAPI nuclear staining (white) effectively illustrating the different morphological cells and layers of the HF (E). Plucked HFs in the telogen (F) and anagen phase (G) are shown, with the anagen HF displaying a well-developed ORS, from which keratinocytes emerge. Panel G shows an overview of an anagen HF (20×), while panel H presents a representative 40× image of the keratinocyte outgrowth, corresponding approximately to the yellow boxed region in panel G. Diagram outlining the cultivation process of HFs. The schematic indicates that keratinocytes typically begin to grow out after 72 h in MEF medium, at which point the medium is switched to EpiLife. Plucked HFs from the parietal region of the scalp are directly placed in MEF medium with supplements. After approximately 72 h, first keratinocytes typically emerge, prompting a change to EpiLife + HKGS supplement. The keratinocytes are maintained in culture until they reach maximum proliferation, at which point they can be used for reprogramming into keratinocyte-derived induced pluripotent stem cells (kiPSCs) (I). Overview of the five research questions addressed in this study (J). Starting from the cultivation of plucked hair follicles, we analyzed (I) from which region the first keratinocytes emerge, (II) the timing of outgrowth, (III) morphological changes during culture, (IV) the layer of origin of the first keratinocytes, and (V) how the protocol can be improved to accelerate outgrowth. Scale bar: 100 µm (B, E), 500 µm (F, G ), 1,000 µm (H).

In this study, we examined several aspects of the keratinocyte outgrowth process from plucked HFs, as summarized in Fig. 1J: From which region of the HF (I), when (II), and from which layer (IV) the first keratinocytes emerge. Additionally, we analysed the morphological changes of the HF (III) during cultivation and made improvements to the culture conditions (V).

In which region do the first keratinocytes appear in the HF?

When generating iPSCs from plucked HF using our well-established protocol, we efficiently produced keratinocytes (Klingenstein et al., 2020). However, we regularly observed donor-specific variations in the integrity of the ORS, the key structure for successful keratinocyte outgrowth. Notably, some patients with rare genetic diseases often display an underdeveloped or less-defined ORS, resulting in the absence of keratinocyte outgrowth (Ahmed et al., 2019; McLean & Moore, 2011). To enhance and improve the outgrowth from more challenging HF samples, we conducted a detailed analysis of the initial keratinocyte outgrowth behaviour and mechanisms in adherent HF cultures, focusing on various aspects of this process. We were interested in determining from which region, when and from which layer the first keratinocytes emerge, as well as the morphological changes in the HF during the cultivation period. To objectively quantify these observations, a custom-designed device was used. This tool was placed directly beneath the culture plates of the HFs, ensuring that the dermal papilla at the distal end of the HF touched the first vertical line of the device. These lines appeared as faint dark marks in the background of the HFs (Fig. 2A). The principle of this tool was to divide the analyzed HFs, which all had very similar lengths, into five equal sections and regions. This enabled measurements to be consistently taken at the same location across various HFs. The small graphic in the upper right corner shows an abstract version of the device (Fig. 2B). Although region division was based on equal-length segments, inter-donor variability in follicle size and shape may limit the anatomical precision of this approach. Using this objective method, we found that most keratinocytes developed from the distal second region of the HF. In total, 36 HFs were analyzed for this experiment, of which 1 HF (2.78%) developed keratinocytes in the first region, 29 HFs (80.55%) in the second region, and 6 HFs (16.67%) in the third region of the device (Fig. 2E). A representative HF is shown in Fig. 2C, where the first keratinocytes were visible in this specific region. The region is characterized by the transition between the hair papilla and the ORS. The magnification of this region, marked with a black arrow, shows the first cells appearing in this area (Fig. 2D).

Figure 2 Localization of first keratinocytes in plucked HF.

To precisely identify the region where initial keratinocyte growth occurs, a device was positioned beneath the HFs for microscopic examination. This device is visible as black lines in the background of the follicles (A). The device allows the HFs to be divided into five equal parts (B). The small graphic in the upper right corner shows an abstract version of the tool, with the lines marking the beginning and end of the HF depicted in black (B). Representative HF where keratinocytes develop from the distal second region of the device (C). In the highlighted magnification, the first cells are marked with a black arrow (D). The percentage of HFs that developed keratinocytes in each region is displayed (E). Most cells (80.55%) were observed in the second region, followed by the third region (16.67%), and the fewest cells (2.78%) in the first region. Number of single, biological experiments per measured region: n = 36 HF. Scale bar: 500 µm (A–C), 400 µm D.

Main results → First keratinocytes mainly grow out in the second region of the distal part of the plucked HF

When do the first keratinocytes appear in the HF?

In addition to the region where the first keratinocytes emerged, we were interested in the time span until the initial appearance of the cells. The protocol mandates a medium change as soon as the first keratinocytes grow out. This is necessary because primary keratinocytes are highly sensitive to calcium and require a specialized medium (EpiLife with supplements) that supports further growth and proliferation (Fujisaki et al., 2018; Bikle, Xie & Tu, 2012; Elsholz et al., 2014). In contrast, HFs require specific culture conditions to ensure that the hair organ, with all its morphologically distinct cell types, adheres to the culture plate and maintains its mitotic activity (McLean & Moore, 2011). Therefore, it would be extremely helpful to know the typical time frame in which the medium change will occur and whether any gender-specific differences can be observed. To answer these questions, we analysed the HFs from four different biological donors (two female, two male). Figures 3A–3E exemplarily shows the emergence of the first keratinocytes from a HF over a 96-h observation period, at 24-h intervals. The HFs were cultured in MEF medium + supplements immediately after plating (0 h) in the tissue culture dish (Fig. 3A) until the first cells appeared (Figs. 3B–3D, 3F). Afterward, the medium was switched to EpiLife + supplements (Fig. 3E). Our results showed that the first cells were observed after 72 h (Fig. 3F, marked with a black arrow). The pie chart summarizes the quantification of the results with respect to appearance of the first keratinocytes (Fig. 3G). In total 24 HFs were analysed for this experiment, of which 1 HF appeared after 96 h (4.17%), 6 HFs (25%) after 24 h, 8 HFs (33.33%) after 72 h and the most keratinocytes appeared after 48 h (n = 9, 37.5%). Overall, the average median time until the first outgrowth across all measured HFs was 48 h. When analysing the data by sex and separating the female and male samples, the graph indicates that the appearance of the first keratinocytes in female samples occurred between 24 and 96 h, with a median of 60 h. The graphic also shows the mean of 56 h and the highest and lowest measured values. Male samples showed the range from 72 h the latest to 24 h the first appearing cells, with the average median of 48 h. The graphic also displays the mean of 48 h and the highest and lowest measured values. From all tested samples, keratinocytes grew out and no significant difference between female and male samples could be detected (Fig. 3H). While the comparison suggests a slight delay in keratinocyte outgrowth in female HFs (median 60 h vs. 48 h in males), the small number of donors (n = 2 per sex) limits the statistical power of this observation. Therefore, these findings should be considered preliminary and descriptive.

Figure 3 Timeline of first appearing keratinocytes from cultivated HF.

The same HF was observed over a period of 96 h (A–E) with the first keratinocytes emerging after 72 h (D). In the magnified image, the initial keratinocytes are marked by a black arrow (F). Pie chart illustrates the duration (h) until the first keratinocytes emerged in 24 analyzed HFs (G). Most of the cells (37.5%) were observed after 48 h, followed by 72 h (33.33%), 24 hs (25%), and the fewest cells (4.17%) after 96 h. The box plot for male HFs (blue) shows a median of 48 h for the appearance of the first cells. The box plot for female samples (orange) indicates a median of 60 h until the first keratinocytes appeared (H). The number of individual biological experiments per measured time point: n = 12 HFs (H). The error bar represents the SEM of the median. Scale bar: 500 µm (A–E), 400 µm (F).

Main results → Overall median time span: 48 h

→ The median time span of first appearing keratinocytes was male HF: 48 h

→ The median time span of first appearing keratinocytes was female HF: 60 h

Morphological changes of plucked HF during cultivation

During our studies, several hundred HFs were analyzed, and it became evident early on that the cultured HF undergo significant morphological changes over time. Figures 4A–4F exemplify these changes in a cultured HF. At the beginning (0 h), the HF remained intact and showed a well-defined structure. There was no visible cell outgrowth, and the follicle retains its original morphology (Fig. 4A). After 24 h, slight cellular activity started around the follicle, with subtle changes in the surrounding area, but no significant cell outgrowth was visible (Fig. 4B). By 48 h, the first keratinocytes became visible around the follicle, particularly in the lower region between the papilla and the ORS (Fig. 4C). After 72 h, the cell growth became more pronounced. The cell density around the HF increased, and the spread of keratinocytes was clearly visible around the structure (Fig. 4D). By 96 h, there was considerable keratinocyte outgrowth, forming a distinct layer of cells spreading from the follicle (Fig. 4E). The cell density further increased after another 24 h, indicating strong proliferation and migration (Fig. 4F). A noticeable increase in the thickness of the ORS was observed, along with the development of an irregular, wavy appearance (Fig. 4G). Within this area, bubble-like structures could be identified, exhibiting a round, swollen morphology (yellow arrows, Fig. 4G). These structures had clear, smooth contours and appeared somewhat transparent or translucent. Their surface was uniform, lacking prominent features, and they stood out distinctly from the surrounding ORS due to their inflated, balloon-like appearance. Based on these morphological characteristics, we propose the term “keratocysts” for these structures. This trend is also observable after 48, 72, and 96 h of incubation. The ORS became increasingly thicker and more irregularly wavy, and the first keratinocytes began to emerge from the ORS area of the “keratocysts”. In contrast, we never observed outgrowth from HFs that did not form such protrusions. Therefore, their appearance serves as an early indicator of successful keratinocyte activation. In practice, we use these morphological changes as a guide: once protrusions are visible, we maintain the HFs in MEF medium for one additional day before switching to EpiLife to support further expansion. This observation may be useful for refining culture decisions and improving efficiency. In all analysed HFs, an increase in the thickness of the ORS was noted in the regions of the hair organ: the proximal area closer to the papilla and directly at the transition between the papilla and the ORS, as well as in a more distal area (Fig. 4H).

Figure 4 Morphological changes of plucked HF during cultivation.

Morphological changes in plated HF after 0 h (A), 24 h (B), 48 h (C), 72 h (D), 96 h (E), and 168 h (F). Over the cultivation period, an increase in the thickness of the ORS is observed in both the distal and proximal regions of the hair organ. These two parameters, along with the change in length between the hair papilla and the ORS (H), were quantified. The graph illustrates the change in the length and width of the ORS relative to directly plated HFs (0 h) for all measured HFs (I). In the negative control, the HF were incubated with water over a period of 96 h (K–O). Quantification shows the change in the length and width of the ORS relative to directly plated HFs (0 h) for all measured HFs (J). A detailed graphical representation of the morphological changes in HF, separated by gender, is provided. The error bar indicates the SEM of the median. Scale bar: 500 µm (A–H, K–O).

To objectively quantify these observations, our self-designed device was employed. The precise division of the HF into equally sized sections allowed for measurements at the same position across different HFs. In total, three different measurements were conducted (Fig. 4H). We could show in all examined HFs (n = 12 per time point), a significant increase in thickness was observed in both the distal and proximal regions of the ORS (Fig. 4I). After only 24 h, an increase in thickness was evident in the proximal and distal areas of the HFs, which continued to rise steadily throughout the duration of the experiments. In contrast, the distance between the papilla and the beginning of the ORS changed only minimally. We further analysed these results by sex and provided a more detailed account of the individual parameters. When comparing the measurements of female HFs (n = 6 per time point, Figs. 5A–5D) with those of male HFs (n = 6 per time point, Figs. 5E–5H), it became clear that the morphological changes in the HFs of both sexes were quite similar. The only noticeable difference was in the distance between the papilla and the ORS, where a significant change in length was observed in the female HFs after 24 and 48 h (Fig. 5D). In parallel, a negative control was conducted (n = 4 per time point), in which the HFs were cultured only with water instead of culture medium (Figs. 4K–4O). Water served as a harsh negative control and consistently induced follicle degeneration, impaired survival and proliferation, as previously shown in Wustner et al. (2022). As expected, there was a decrease in the thickness of the root sheath in both the proximal and distal areas (Fig. 4J). The length growth between the papilla and the ORS initially showed an increase but eventually plateaued at a consistent level. Furthermore, no outgrowth of keratinocytes was observed in these experimental series. However, a notable darkening of the HF was evident after 48 to 72 h of incubation. A precise differentiation and distinction of the hair shaft, as well as the inner and ORS, was almost impossible to achieve microscopically. These observations suggest that the process of HF degeneration begins after 48 h, but no later than 72 h. This could be attributed to the lack of essential nutrients during incubation with water, which are present in the MEF base medium to maintain homeostasis.

Figure 5 Quantitative analysis of morphological changes in hair follicle regions.

(A) shows the measured female HF with proximal (B), distal (C), and the distance between the papilla and ORS (D). Graph E shows the measured male HF with proximal (F), distal (G), and the distance between the papilla and ORS (H). Number of single, biological experiments per measured time point: n = 6 (A –H). TTEST p < 0.05 *, p < 0.01 **, p < 0.001 ***. The error bar indicates the SEM of the median.

Main results → Morphological changes during cultivation include thickness growth of the HF in the distal and proximal regions of the ORS, along with length growth

→ “Keratocysts”, bubble-like structures in the area of the ORS, are visible after just 24 h

From which layer of the HF do the first keratinocytes emerge?

When plucking human HFs to obtain keratinocytes, we observed typical morphological changes, including the formation of bubble-like structures along the hair root. Our observations showed that keratinocytes only emerge from the HF when such structures are present. This raised the question from which specific layer of the hair organ these initial keratinocytes originate from and whether they can be associated with a defined cell population. To address this question, we conducted a comprehensive analysis of HFs, focusing on the expression of keratins in longitudinal and cross-sections of HFs, as well as during the early stages of cell growth. Figure 6 shows longitudinal (Figs. 6A–6D) and cross-sections (Figs. 6E–6H) of plucked HFs. The structure of the outer root area and the layering of cells within the HF are clearly visible by co-staining with DAPI (blue). Stainings were performed for Keratin 5, 6/75, 14, and 15 (red). Keratins K5, 14, and 15 specifically mark the outermost basal cell layer of the ORS. In addition to this layer, the K6/75 marker, which detects both Keratin 6 and Keratin 75, also marks the companion layer of the inner root sheath (IRS).

Figure 6 Keratin expression in plucked hair follicles.

Expression of keratin markers K5, K6/75, K14, and K15 (red) in longitudinal (A–D) and cross-sections (E–H) of plucked HFs. Scale bar: 100 µm (A–H).

The brightfield image shows the bubble-like keratinocytes located at the edge of the ORS (Fig. 7A, marked with orange arrows), which are positive for Keratin 14 (Fig. 7B). In another brightfield image, the initial outgrowth of keratinocytes from the plucked HF is visible (Fig. 7C). There was a population of keratinocytes which are exclusively positive for Keratin 5 (Fig. 7D, marked with yellow arrows) and cells which are double-positive for Keratin 5 and Keratin 15 (Fig. 7E, marked with white arrows). Another brightfield image shows the structure of the HF at a later stage (Fig. 7F). The central part of the follicle is visible, with an enlargement in the form of thickening of the ORS in the distal and proximal regions. Some keratinocytes had already grown out and proliferated. Detailed immunofluorescence images of the same HF region stained for K5 and for K15 (Figs. 7G, 7G′) are shown, including corresponding magnified sections. Keratinocyte populations exist which are positive only for K15 (Fig. 7H, marked with red arrows), as well as cells which are double-positive for K5 and K15 (Fig. 7H, marked with magenta arrows). The IF stainings in Figs. 8A–8F show the staining of keratin markers (K5, K6, K6/75, K10, K14, K15) in adherently growing, proliferating keratinocytes. The markers previously detected in the flattened HF were also positive. Notably, Keratin 10 (Fig. 8D) was mainly expressed in adherently growing keratinocytes, not in the flattened HF or in the early outgrowing cells. Quantification of the relative gene expression of Keratin 5, K10, and K14 in plucked HFs and adherently growing keratinocytes was shown in addition (Fig. 8G). The graph shows that in the HF, only Keratins 5 and 14 were expressed, whereas in proliferating keratinocytes, all three tested markers were expressed.

Figure 7 Keratin expression in first outgrowing and adherent keratinocytes.

Brightfield image (A) of the bubble-like keratinocytes in the area of the ORS, with an IF image (B) of the same region stained for K14 (green). Brightfield image (C) showing the first outgrowing keratinocytes in the ORS area, with corresponding IF images for K5 (D) and K15 (E). Cells positive only for K5 are marked with orange arrows, and double- positive K5 and K15 cells are marked with white arrows Brightfield image (F) of a HF at a later culture stage, showing thickening of the ORS in distal and proximal regions, with proliferating keratinocytes visible near the follicle. Immunofluorescence staining for K5 and K15 in the same HF region (G). In the magnified section (H), magenta arrows mark K5/K15 double-positive keratinocytes, red arrows indicate cells positive for K15 only. Scale bar: 500 µm (F, G), 250 µm (A–E), 100 µm (H).

Figure 8 Keratin expression in adherent keratinocytes.

IF stainings of individual keratins K5, K6, K6/75, K10, K14, and K15 (A–F) in adherently growing, proliferating keratinocytes. Quantification of relative keratin expression for K5, K10, and K14 in plucked HFs (G, red bars) and adherently growing keratinocytes (G, magenta bars). Notably, K10 is expressed only in mature keratinocytes at both RNA and protein levels. Scale bar: 100 µm (A–F).

Based on the keratin expression observed in the HFs, in the initially outgrowing cells, and in the proliferating keratinocytes, we can infer that these cells share characteristics with the basal cell layer of the ORS. While this pattern is consistent with an ORS origin, it does not conclusively determine the precise source layer.

Main results → First appearing keratinocytes express the same makers as cells of the ORS

→ K10 is not expressed in plated HF and first outgrowing keratinocytes

Improvement of the HF cultivation protocol

In our research group, we use primary keratinocytes from plucked HF for reprogramming into keratinocyte-induced pluripotent stem cells (kiPSCs). Over the years of working with HF from many donors, we have observed significant individual differences in the quality of the HFs, particularly in the ORS. This is especially noticeable in patients with genetic disorders. To increase the likelihood of outgrowth and improve the yield of keratinocytes, we next aimed to optimize the protocol for generating keratinocytes. The main goal was to shorten the time until the first outgrowth of keratinocytes to expedite the overall reprogramming process. The objective is faster the outgrow regardless of the donor, and therefore to develop a uniform and standardized protocol. To achieve this goal, the addition of various growth factors and small molecules to the culture medium was tested. All tested compounds target signaling pathways associated with increased cellular proliferation, including TGF-β, Notch, and EGFR signaling. The following molecules were added to the standard MEF medium: 10 µg/ml A83-01, 10 µg/ml DAPT (N-[N-(3,5-Difluorophenacetyl)-L-alanyl]-S-phenylglycine t-butyl ester), 20 ng/ml EGF (epidermal growth factor), 20 ng/ml IGF-1 (insulin-like growth factor-1), and 10 µg/ml SB431542. As controls, we included both MEF medium without any supplements (MEF) and standard MEF medium containing the usual additives (MEF++). Once the first keratinocytes became visible, the timepoint was recorded and analysed. The data from a total of 120 HFs revealed a robust reduction in outgrowth time when using specific additives (Fig. 9A). Among all tested compounds, DAPT, a Notch pathway inhibitor, led to the most pronounced acceleration of outgrowth, with a mean of 35 ± 21 h (vs. 69 ± 23 h in MEF++ control, p < 0.001). IGF-1 also significantly reduced the time to outgrowth (40 ± 21 h) compared to both controls. A83-01, EGF, and SB431542 led to moderate reductions in mean outgrowth times (48–53 h), with statistically significant differences for A83-01 and EGF compared to MEF++ (p < 0.05). These effects likely result from partial stimulation of keratinocyte migration and proliferation, with DAPT having the most direct and potent impact. Sex-stratified analysis further revealed donor-specific variability across treatments. No significant differences between male and female donors were observed for any of the tested conditions (Fig. 9B). Overall, the results support DAPT as the most effective compound for accelerating keratinocyte outgrowth from plucked HFs, with potential for improving reprogramming workflows and enhancing culture robustness in challenging donor samples.

Figure 9 Cell culture supplements to shorten the time to the first outgrowth of keratinocytes.

(A) Median time (white numbers) until first visible keratinocyte outgrowth from plucked human hair follicles cultured in standard MEF medium with supplements (MEF++, 69 h, n = 35) or MEF medium alone (MEF, 65 h, n = 10) as control, and in the presence of five different additives: A83-01 (51 h, n = 15), DAPT (35 h, n = 19), EGF (53 h, n = 14), IGF-1 (40 h, n = 12), and SB431542 ( 48 h, n = 15). (B) Bar diagram showing individual biological measurements for each supplement, separated by donor sex: male HFs (blue), female HFs (orange) and median time (white numbers). A83-01 (n: m = 7, f = 8), DAPT (n: m = 9, f = 10), EGF (n: m = 10, f = 4), IGF-1 n: (m = 7, f = 5), SB431542 (n: m = 7, f = 8), MEF (n: m = 6, f = 4), MEF++ (n: m = 19, f = 16). Statistical analysis was performed using unpaired two-tailed TTEST *p < 0.05, **p < 0.01, ***p < 0.001, n: Biological experiment per group. Each bar represents the mean ± SD of individual biological replicates.

Main results

→ The addition of DAPT significantly reduces the time until first keratinocytes grow out (35 h)

Discussion

Primary keratinocytes are particularly important for generating iPSCs, as they represent an easily accessible source that can be obtained from virtually any individual by plucking HFs. This facilitates their application in personalized medicine since they enable the generation of patient-specific stem cells that precisely mirror the individual’s genetic makeup. These iPSCs can be differentiated into various cell types to model specific diseases or to develop personalized therapeutic approaches. As part of our research, we have collected hair samples from a variety of healthy and diseased individuals, cultured them, and reprogrammed them into iPSCs. In these cases, we observed notable differences: keratinocytes often took significantly longer to grow out of the hair follicles, or growth was unsuccessful altogether. To better understand the structural consequences of plucking prior to analyzing the region and timing of keratinocyte outgrowth, we included a direct comparison between skin-embedded and plucked HFs in Figs. 1B, 1C. These images show that while the arrector pili muscle and dermal sheath are consistently lost during plucking, the ORS remains structurally intact and maintains its characteristic layered morphology. This is evident in both the histological section of a skin-embedded HF and the inverted DAPI staining of a plucked HF. Only in rare cases (~1 in 40 HFs), the dermal sheath is still attached to the plucked follicle. The identification of telogen HFs in our study was based on morphological appearance only. Since the success of plucking varies strongly between individuals, the frequency of obtaining telogen HFs can range from <5% to nearly 90%. While the morphological criteria are well established, variability in plucking technique and anatomical differences may influence the observed structure. We acknowledge that our classification may include variability due to the absence of histological confirmation. To optimize the culture conditions specifically for these HFs and to speed up and improve the overall process, we analyzed and optimized the entire cultivation process of HFs from healthy control subjects in detail. We made exciting observations which are scientifically presented and evaluated for the first time in this research work.

Where and when do first keratinocytes appear?

The results from analyzing 36 HFs revealed exciting findings regarding when and where the first keratinocytes emerge. We were able to show that most cells (80.55%) grow out from the second fifth of the whole plucked hair organ. This area is characterized by the transition between the papilla shaft and the ORS and may be explained by the fact that this region exhibits high cellular turnover, including stem cells (Lin et al., 2020). These stem cells are highly proliferative and play a central role in the regeneration and growth of HFs (Hu et al., 2021). Therefore, this region may contain a high concentration of cells ready to differentiate into keratinocytes. When the HF is plated and cultured in an appropriate medium, these stem cells may be activated, promoting the outgrowth of keratinocytes in this region. To define this region systematically across donors, we divided each plucked HF into five equal-length segments using a standardized measuring device. While this allowed for consistent positional mapping of outgrowth events, it does not account for inter-follicular anatomical variability. The segmentation is purely geometric and may not correspond to histologically defined zones. Future studies using morphological landmarks or molecular markers could refine this approach and provide greater biological resolution. We observed this initial outgrowth in 24 examined HFs after an average of 48 h (37.5%). More than a third of the HFs (33.3%) showed an outgrowth after 72 h. When analyzing the data by gender, we found a tendency for later outgrowth (52 h) in female HFs. A possible explanation for these observations may lie in the cell activation and subsequent migration of the first keratinocytes. After plucking the HF and transferring it into culture, the keratinocytes in the hair organ may need some time to become activated. The outgrowing cells likely originate from the area between the papilla region and the ORS, where a stem cell niche is thought to exist (Zhang & Chen, 2024; Chen et al., 2020; Rompolas & Greco, 2014). These cells must first leave their quiescent state before becoming active. This process is initiated by signals from the environment, such as growth factors or cytokines, added to the MEF culture medium. It is also possible that these signals are additionally released by the dermal papilla cells, which remain intact after the HF was plucked. This activation is crucial to initiate cell migration and the outgrowth of keratinocytes, which takes about 48 h, as the necessary biochemical processes require time to develop. It should be noted that the observed differences between male and female samples are based on a limited number of donors (n = 2 per sex) and should therefore be interpreted with caution. Further studies with larger donor cohorts are needed to determine whether sex-specific differences in keratinocyte outgrowth exist.

How does the morphology of the HF change during cultivation?

We observed that during the initial phase of HF cultivation, significant morphological changes occur, not only in the region where the ORS begins but also in other measured parameters. Over a period of 96 h, we monitored a total of 60 HFs under standard culture conditions and 20 HFs under control conditions (water). We found that the morphology of the HF changed markedly. Immediately after plating, the HF remained intact with no visible cellular changes. After 24 h, early signs of cellular activity appeared, although without significant cell growth. After 48 h, the first keratinocytes became visible, especially in the region between the papilla and the ORS. After an additional 24 h, cell growth intensifies, and more keratinocytes spread around the follicle. These observations were quantified by a significant increase in thickness in both the distal and proximal regions of the HF. Additionally, a notable lengthening of female HFs was observed at the start of plating. One particularly striking morphological change was the formation of bubble-like structures (“keratocysts”) at the edge of the ORS as early as 24 h after plating. These bubbles may be early signs of cellular protrusions, where keratinocytes begin to migrate from the settled HF. These protrusions likely form as the cell membrane material pushes forward before the keratinocytes fully detach, migrate out of the HF, and proliferate. This could represent a preparatory phase where the cells in the ORS alter their morphology to make room for the migration and growth of the first keratinocytes. These bubble-like structures, the “keratocysts”, may signify a transitional phase before the cells further differentiate and release keratinocytes.

Which layer of the HF do the keratinocytes emerge from?

The question arose whether it is possible to trace from which layer of the HF the first cells migrate. The changing morphology of the HF may provide the first clues. The images clearly demonstrated the emergence of cells from the hair follicle, with the first cell colonies forming. These colonies were observed directly along the edge of the basal layer of the ORS. This provides a clear indication that the ORS plays a crucial role in determining which cells migrate and proliferate first. In addition to these cellular observations, we consistently documented an expansion of distal and proximal HF regions during culture. Although the underlying mechanism remains unclear, it may reflect active structural remodeling within the HF or passive mechanical displacement caused by surrounding keratinocyte outgrowth. We did not assess time points beyond 96 h or measure hair shaft elongation. Nevertheless, we considered it important to quantify and include this reproducible morphological change as part of the culture dynamics of plucked HFs. Complementing these results with the expression of the analyzed keratin markers, particularly K5, K14, and K15. The staining for K5 and K14, which are specific markers for the basal cell layer of the ORS, showed that the structures and cells in the early stages of outgrowth expressed these markers. The cells continued to express these markers at later time points, both during the proliferation phase of keratinocytes from the plated HF and in keratinocyte culture. These observations were confirmed at both the RNA levels. Notably, keratin 10 was expressed exclusively in keratinocytes, not in the plated HF or in the early cells. This supports previous analyses that describe K10 as a marker for mature, differentiated keratinocytes (Dmello et al., 2019; Hazrati et al., 2024). Thus, the first outgrowing cells are still young, proliferative, mitotically active, and not yet fully differentiated keratinocytes. The expression of K5, K14, and K15 markers remained consistent both in the very early stages of outgrowth and in the later proliferation phases. Although the marker profile of the outgrowing keratinocytes is compatible with an ORS phenotype, we cannot rule out that cells from inner HF layers may have de-differentiated during culture. This suggests that the cells indeed grow out from the ORS, progressing through stages of “keratocysts” and early undifferentiated keratinocytes to more mature, proliferating cells. This supports our observations that successful keratinocyte cultivation and outgrowth are only possible in the anagen phase of the HF, where a visible ORS is present. Table 1 summarizes the expression of the analyzed keratins at both the RNA and protein level.

Table 1 Summary of keratin expression.

	Protein expression	RNA expression	
Keratin	Plated HF	First outgrowing Keratinocytes	Keratinocytes	Plated HF	Keratinocytes	
K5	+++ ORS	+++ ORS	+++	+++	++	
K6	n.t.	n.t.	+++	n.t.	++	
K6/75	+++ ORS	n.t.	+++	n.t.	n.t.	
K10	–	–	+++	−/+	+++	
K14	+++ ORS	+++ ORS	+++	+++	n.t.	
K15	+++ ORS	+++ ORS	+++	n.t.	n.t.	
Note:

The keratins K5, K6, K6/75, K10, K14, and K15 were tested in plated HFs, plated HFs with first outgrowing keratinocytes, and in keratinocyte populations only, at both RNA and protein levels. Some markers were not tested in certain conditions (n.t.). “+++”: very high, “++”: high, “+”: moderate, “−”: no expression.

Improvement of outgrowth speed

In the final step, we sought ways to reduce the time it takes for keratinocytes to first emerge. Our goal was to identify a cell culture supplement that shortens the entire process from hair plucking to reprogramming. This is particularly important for HFs from patients where the process tends to be slower and more challenging. Additionally, we aimed to improve predictability in determining the optimal time for medium changes. By doing so, we aimed to develop a faster and more well-defined protocol. For this, we analyzed a total of 120 HFs and tested five growth factors or small molecules in comparison to the control medium without additional supplements. DAPT emerged as the clear favorite. By adding this Notch inhibitor, the time until the first keratinocyte outgrowth was reduced from an average of 48 to 35 h. The positive effects of DAPT on keratinocyte cultivation have already been described in previous studies (Tadeu et al., 2015; Ichida et al., 2014). Previous studies have shown that adding the γ-secretase inhibitor, which blocks the Notch signaling cascade, is highly effective in stimulating processes necessary for the migration and proliferation of keratinocytes from plucked HFs. Notch signals play a crucial role in maintaining undifferentiated cell states as well as regulating cell proliferation and differentiation (Grotek, Wehner & Weidinger, 2013; Gioftsidi, Relaix & Mourikis, 2022). By inhibiting these signals, DAPT may accelerate keratinocyte differentiation, potentially allowing the cells to transit more rapidly from a proliferative state to mature keratinocytes, facilitating the first visible outgrowth. Since Notch signals often regulate cell division, their inhibition by DAPT may also lead to faster cell proliferation and increased cell migration (Alhashem et al., 2022). These effects may enable keratinocytes to migrate out of the HF more quickly and spread on the culture surface, thereby speeding up the overall cultivation process. The addition of DAPT to the MEF basal medium thus offers the significant advantage of making cell culture faster, more efficient, and more predictable—especially for difficult patient samples.

Conclusion

Despite the detailed morphological insights provided by our ex vivo system, we acknowledge that plucked human hair follicles are studied outside their native skin environment. This limits our ability to assess the influence of surrounding factors such as immune cells, vasculature, fibroblasts, and epidermal–dermal signaling. These components are known to contribute to the autoregulatory network of the skin and modulate hair follicle behavior in vivo (Sasaki, 2019; Zhang & Chen, 2024). Therefore, while our findings provide a valuable foundation for standardized HF culture and keratinocyte harvesting, they do not fully capture the complex interactions present in situ. With the results of this study, we hope to make a substantial contribution to the understanding of the entire process from HF extraction to the reprogramming of keratinocytes. This study is also intended to be a valuable resource for scientists who are new to the field of reprogramming. Our research findings aim to serve as a reference, showing how HFs morphologically change under optimal physiological conditions during culture. Additionally, they will provide insight into which HFs were suitable for culture and what their characteristic appearance should be. Our findings may contribute to more standardized and efficient protocols for iPSC generation from patient-derived keratinocytes, which is particularly relevant in the context of personalized regenerative medicine.

Supplemental Information

Supplemental Information 1 Raw data.

Supplemental Information 2 HF donors with different sex and age.

Supplemental Information 3 List of different cell culture media and supplements used for HF cultivation.

Supplemental Information 4 List of cell culture supplements.

Supplemental Information 5 List of blocking solution and antibodies used for immunostaining.

We are deeply grateful to the body donors whose selfless contributions made this research possible.

Additional Information and Declarations

Competing Interests

The authors declare that they have no competing interests.

Author Contributions

Stefanie Klingenstein conceived and designed the experiments, performed the experiments, analyzed the data, prepared figures and/or tables, authored or reviewed drafts of the article, and approved the final draft.

Judith Boenke performed the experiments, analyzed the data, prepared figures and/or tables, and approved the final draft.

Lisa-Sophie Wüstner performed the experiments, prepared figures and/or tables, and approved the final draft.

Stefan Liebau conceived and designed the experiments, authored or reviewed drafts of the article, and approved the final draft.

Moritz Klingenstein conceived and designed the experiments, authored or reviewed drafts of the article, and approved the final draft.

Human Ethics

The following information was supplied relating to ethical approvals (i.e., approving body and any reference numbers):

The study was conducted in accordance with the Declaration of Helsinki and approved by the Ethics Committee at the Medical Department of the University Tübingen (Project number: 638/2013BO01, date of approval: 31.3.2014).

Data Availability

The following information was supplied regarding data availability:

The raw data are available in the Supplemental Files.

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
