# Peer review of "Detailed morphological characterization and improvement of keratinocyte outgrowth from plucked human hair follicle"

_PeerJ, doi:10.7717/peerj.20214_

## Round 0.1 · original submission · Major Revisions

·

Basic reporting

This is a technically sound paper, and I do not have a major critique here. My main critique relates to innovation and significance in human biology. The references are limited for the field of hair biology.
The introduction is short and does discuss recent concepts in skin biology, and the hair follicle is part of this system. Therefore, proper context would be appreciated.
The discussion of limitations is suggested not only in technical issues but also in conceptual ones in the context of the interaction of hair follicle and follicular keratinocytes with cutaneous auto-regulatory mechanisms.

Experimental design

This is a technically sound paper, and I do not have a major critique here.
The experimental design appears to be appropriate.

Validity of the findings

-

Reviewer 2 ·

Basic reporting

While the manuscript presents a potentially valuable study on the outgrowth of keratinocytes from plucked human hair follicles and its implications for regenerative medicine, several critical issues limit its current suitability for publication.

Although the abstract highlights the clinical relevance of optimizing culture conditions, the manuscript as a whole suffers from issues of clarity, methodological rigor, and data presentation. Specifically, there are multiple instances of unclear figure legends and missing references in the main text (Figure 1D), inconsistent or missing scale bars, and a lack of proper statistical analysis. The use of qualitative scoring systems (“+++”) in place of quantitative values reduces the interpretability of key results (Table 1). Moreover, grammatical errors and imprecise descriptions throughout the manuscript indicate a need for comprehensive English language revision.

Of particular concern is the experimental design regarding sex-based comparisons. With only two male and two female donors, any conclusions drawn about sex-specific differences lack sufficient statistical power and validity, raising questions about the robustness of the findings.

Key points
1. Kindly correct the term "Katagen" to "Catagen" in line 180. The same correction is also needed in Figure 1A.
2. Figure 1D is not referenced in the main text.
3. For Figure 1G, a mention of the time point for the keratinocytes in culture is needed.
4. Figure 1H is unclear. Please consider revising the illustration to more clearly convey the meaning of the arrows within the hair follicle and the connection between the circled elements.
5. In Figure 1B, the legend states a scale of 100 μm, but the actual scale bar is missing from the image.
6. Line 127 (“The hair shaft, the visible hair structure extending beyond the skin”) is grammatically incorrect. A general proofreading of the manuscript is recommended to improve the clarity and accuracy of the English.
7 The yellow box in Figure 1E does not accurately correspond in scale or position to the magnified region shown in Figure 1F. This should be corrected for consistency and clarity.
8. Figure 2E, 3F – There is an incorrect use of a comma in the decimal number "2,78%." Please revise it to "2.78%" for proper formatting.
9. Figure 4O – The term "negativ control" appears to be a typographical error. It should be corrected to "negative control." Please ensure consistent and accurate usage throughout the manuscript.
10. It is recommended to add a Statistical Analysis section to the Materials and Methods for completeness and clarity.
11. Scale bars should be added to the figures throughout the manuscript for clarity and consistency.

Experimental design

The study utilizes hair follicles from four different biological donors (2 female, 2 male). However, with only two individuals per sex, it is questionable whether sex-based comparisons yield meaningful conclusions. A critical reflection on the experimental design is warranted, particularly regarding the validity and statistical power of sex-specific analyses with such a limited sample size.

Validity of the findings

In Table 1, the use of qualitative symbols such as "+++", "++", and "+" to indicate expression levels may lead to an ambiguous interpretation. It is recommended to replace these with precise numerical values, where possible, to improve clarity and allow for more accurate comparison.

Reviewer 3 ·

Basic reporting

-

Experimental design

1. Cell culture of outer root sheath cells derived from human hair follicles is already a very established method (Weterings et al., 1981; Arase et al., 1991). The authors should acknowledge this in the introduction of their paper and clearly delineate what additional information they are contributing.

2. The authors show light field images of plucked “anagen” and “telogen” hair based on the presence or absence of the ORS (Fig. 1D and E). What is the frequency of obtaining these “telogen” hairs with plucking? The hair shaft shown as "telogen" does not have the typical "club hair" morphology and could be an anagen hair without the ORS. Could the authors be sure that this was telogen hair without being able to verify with histology of the remaining follicle structure in the skin? In other words, could this difference in morphology of plucked hair be caused by the plucking method? It would be nice if the authors could add a paragraph discussing the plucking skills to make the manuscript more comprehensive.

3. The authors started culturing the human hair follicles in MEF medium with supplements. What is the purpose of using the MEF medium at the beginning of the culture? What is observed when the Epilife medium is used from the beginning? I would like to see a comparison of the efficiencies of the two media.

4. The authors essentially use a ruler to split the hair follicle into different regions. However, this is an inconsistent method when considering variability in hair follicle length and morphology across different donors. In figures 2B and 2C, for example, the delineation of the “first region” seems to differ - in 2B, the cutoff seems to be further away from the hair bulb, while in 2C it is closer to the hair bulb. Without a clear justification and explanation for determining these regions, it becomes difficult to extrapolate any downstream results using these parameters. Morphological or molecular cutoffs may be more appropriate to standardize the respective regions of the hair follicle being analyzed.

5. The authors compared outgrowth timing between male and female donors. However, there were only two donors in each sex group. To make the data meaningful, the authors need to increase the sample size of each sex group. In addition to sex, what about the influence of age on cell growth? It would be interesting if the authors could study plucked hair from donors across different age groups.

6. The authors attempt to make a claim about the time it takes for the first cultured cells to appear. However, to fully determine this, they should utilize a more rigorous statistical method (i.e., mean with standard deviation) for assessing the expected outgrowth time. This is particularly important given that they subsequently attempt to shorten the time to form the first outgrowths by manipulating culture conditions.

7. Figure 4: The authors document the morphological changes of the attached HFs over culture. However, what is the biological significance of these changes? In other words, the authors need to address the relationship between observing these protrusions from hair follicles earlier and the cell outgrowth later. The first protrusion seems to occur at about 24 hrs at two different locations: the Distal and proximal sections of the hair. Was this always the case across multiple donors? Are these protrusions the sites where the outgrowth would occur later? Have the authors ever seen an HF form these protrusions but without cell outgrowth, or cell outgrowth without forming these bulges? If there is indeed a positive correlation, the protrusions at 24 hrs could be used as a guide to help the researchers improve culture efficiency by either continuing or terminating the culture at this stage. The authors should address this point.

8. It would also be interesting to address the nature of these protrusions. Are they a result of active cell proliferation within the HF ORS, meaning the cells proliferated within the ORS first and then finally moved out to the culture dish when the integrity of the ORS was compromised? A histology of these bulging structures and a Ki67 staining would be useful to help answer these questions.

9. Culturing the HFs in water as a negative control does not make sense. Water is hypoosmotic and kills the cells. The HFs should be put in PBS at a minimum, or a basic medium (such as the basic MEF medium without supplements), which maintains cell survival but does not support outgrowth.

10. Figure 5 shows a continuous trend of expansion of the distal and the proximal region of the hair follicles in culture. Will this trend continue after 96 hrs? And again, what is the biological significance of this observation? The authors showed that the distance between the DP and the ORS increased as well (at least in the male donors). Does this mean that the HFs grew as an organ, or was this due to the mechanical stretch of the hair by the outgrowth of cells surrounding the hair follicles? A measurement of the hair shaft length (to see if it elongated) could provide some insight into this question.

11. The authors showed that, as expected, the ORS expresses basal keratinocyte markers such as K5, K14, and K15. They also showed that the outgrowth cells express these markers as well. They therefore claimed the outgrowth cells originated from the ORS. Although this is most likely the case, expressing the same markers is not direct evidence to make this claim. Arguably, some of the cells could have originated from the inner layers of the hair follicles and were de-differentiated to express these basal keratinocyte markers in culture. The authors should rephrase their claim here, simply stating that the outgrowth cells express basal keratinocyte markers in culture.

12. It was not clear from the text whether the authors were performing double staining of K5 and K15 in Figures 7G and H. These two markers were both shown as green fluorescence on identical sections, which was a bit confusing. If the authors indeed performed double staining (as suggested by the identical DAPI images for both K5 and K15), a triple color overlap (K5/K15/DAPI) would clearly demonstrate the single and double positive cells. In addition, the K15 staining in 7H seems to be non-specific as all the layers of the HF stained positively.

13. Figure 8: The authors showed that some of the outgrowth cells also express K10, a differentiated keratinocyte marker. When do the cells start to express K10 in culture? What was the percentage of cells that became K10+? Are they located at the periphery of the cell outgrowth area? It would be more informative to perform a double staining of K10 with K5 or K14 to show the cell distributions.

14. The authors tested several additives to the MEF medium to see if any of them could have shortened the initial cell outgrowth time. The results would have been interesting, but the way they described how the experiments were done raised a lot of questions. They stated that the test molecules were added to “standard MEF medium,” which in their previous descriptions means “basic MEF + supplements”. However, their control HFs were plated in “basic MEF medium without any additives”. Does this mean “basic MEF without its own supplements”? If so, this is not a fair comparison. Either way, the authors should clarify. If the authors indeed used basic MEF without its supplements as a control, it is not surprising that every additive they tested (added to the standard MEF with supplements) showed a significant improvement over the control.

15. On a separate note, since the authors tested molecules impacting several different pathways (such as Notch, TGF-b, and IGF), is there an additive effect of combining any of these molecules/growth factors?

Validity of the findings

See above. The number of samples is generally too low to draw conclusions.

---

## Round 0.2 · accepted · Accept

Thank you for the careful consideration of the issues raised at first review. I am sorry for the delay in getting a decision to you, but I confirm I am satisfied with your responses.